# In Vitro and In Vivo Anti-Inflammatory Effects of Polyphyllin VII through Downregulating MAPK and NF-κB Pathways

**DOI:** 10.3390/molecules24050875

**Published:** 2019-03-01

**Authors:** Chao Zhang, Chaoying Li, Xuejing Jia, Kai Wang, Yanbei Tu, Rongchun Wang, Kechun Liu, Tao Lu, Chengwei He

**Affiliations:** 1School of Life Sciences, Beijing University of Chinese Medicine, Beijing 100029, China; zhangchao_0213@126.com; 2State Key Laboratory of Quality Research in Chinese Medicine, Institute of Chinese Medical Sciences, University of Macau, Macao 999078, China; jiaxjsicau@gmail.com (X.J.); wangkai6648@gmail.com (K.W.); tuyanbei1992@163.com (Y.T.); 3School of Pharmaceutical Sciences, Changchun University of Chinese Medicine, Changchun 130117, China; chaoying_li@126.com; 4Key Laboratory for Drug Screening Technology of Shandong Academy of Sciences, Shandong Provincial Key Laboratory for Biosensor, Biology Institute of Shandong Academy of Sciences, Jinan 250014, China; 1wangrongchun@163.com (R.W.); hliukch@sdas.org (K.L.)

**Keywords:** polyphyllin VII, anti-inflammation, macrophage, zebrafish, mice

## Abstract

*Background*: Polyphyllin VII (PP7), a steroidal saponin from *Paris polyphylla*, has been found to exert strong anticancer activity. Little is known about the anti-inflammatory property of PP7. In this study, the anti-inflammatory activity and its underlying mechanisms of PP7 were evaluated in lipopolysaccharide (LPS)-stimulated RAW264.7 cells and in multiple animal models. *Methods*: The content of nitric oxide (NO) was determined by spectrophotometry. The levels of prostaglandin E2 (PGE_2_) and cytokines were measured by enzyme-linked immunosorbent assay (ELISA) assay. The mRNA expression of pro-inflammatory genes was determined by qPCR. The total and phosphorylated protein levels were examined by Western blotting. The in vivo anti-inflammatory activities were evaluated by using mouse and zebrafish models. *Results*: PP7 reduced the production of NO and PGE_2_ and the protein and mRNA expressions of pro-inflammatory cytokines (*TNF-α*, *IL-1β*, and *IL-6*) and enzymes (inducible NO synthase [*iNOS*], cyclooxygenase-2 [*COX-2*], and Matrix metalloproteinase-9 [*MMP-9*]) in LPS-induced RAW264.7 cells by suppressing the NF-κB and MAPKs pathways. Notably, PP7 markedly inhibited xylene-induced ear edema and cotton pellet-induced granuloma formation in mice and suppressed LPS and CuSO_4_-induced inflammation and toxicity in zebrafish embryos. *Conclusion*: This study demonstrates that PP7 exerts strong anti-inflammatory activities in multiple in vitro and in vivo models and suggests that PP7 is a potential novel therapeutic agent for inflammatory diseases.

## 1. Introduction

Inflammation, a physiological response to protect the body from infection or tissue injury, plays a critical role in chronic diseases and various human cancers [1]. Macrophages are major immune cells in the innate immune system and play central roles in inflammation and directly counteract harmful external stimuli. Activated macrophage plays key roles in host defenses against infection with pathogens, to initiate phagocytic activities and promote inflammatory responses via producing various inflammatory factors, e.g., nitric oxide (NO) and prostaglandin E2 (PGE_2_); pro-inflammatory mediators, e.g., nitrogen species, inducible NO synthase (iNOS), cyclooxygenase-2 (COX-2), metalloproteinases; and pro-inflammatory cytokines, e.g., tumor necrosis factor-alpha (TNF-α), interleukin-1beta (IL-1β), and IL-6, that recruit additional immune cells to the sites of infection or tissue injury [2,3,4]. Overproduction of the inflammation-related mediators and cytokines by activated macrophages has been implicated in the pathophysiology of many autoimmune diseases and inflammatory disorders [5,6,7]. Lipopolysaccharide (LPS), an endotoxin originally derived from the cell walls of Gram-negative bacteria, is a potent macrophage activator [8]. When macrophages stimulated by LPS, the expressions of these mediators and cytokines are increased and regulated by the activation of the transcription factor nuclear factor-κB (NF-κB), which is critically involved in the pathogenesis of many inflammatory diseases and can be activated via mitogen activated protein kinases (MAPKs), including extracellular signal-regulated kinases (ERKs), c-Jun NH_2_-terminal kinases (JNKs), and p38-MAPK (p38) [9,10]. Matrix metalloproteinase-9 (MMP-9), one of the MMPs secreted by macrophages, regulates leukocyte migration in inflammatory diseases [11]. Lipopolysaccharide regulates *MMP-9* expression through the activation of MAPK and NF-κB signaling pathways in macrophages [10,12]. Therefore, treatments targeting NF-κB, MAPKs, and MMP-9 may have potential therapeutic advantages for inflammatory diseases.

The rhizome of *Paris polyphylla* var. *yunnanensis*, or called *Rhizoma Paridis*, mainly contains steroidal saponins, especially pennogenyl and diosgenyl saponins [13]. Several steroidal saponins with significantly biological activities have been isolated and identified from *Rhizoma Paridis*. Polyphyllin VII (PP7), one of the steroidal saponins from *Rhizoma Paridis*, has been demonstrated to have strong anticancer activity [14,15]. However, the anti-inflammatory activity and its molecular mechanisms are still unknown. In this study, we show for the first time that PP7 significantly suppresses the LPS-induced NO and PGE_2_ production and the protein and mRNA expression of *iNOS* and *COX-2* as well as the pro-inflammatory cytokines and MMP-9 in RAW264.7 cells by inhibiting NF-κB and MAPKs pathways.

Zebrafish (*Danio rerio*), a freshwater tropical fish, has emerged as a useful vertebrate model organism for a variety of inflammation studies because its innate and acquired immune systems are very similar to the mammalian immune system, and the transparency of zebrafish embryos and larvae allows non-invasive and dynamic imaging of the inflammation in vivo [16,17,18,19,20]. Neutrophils are the most abundant leukocytes in adult zebrafish, and the *BACmpx::GFP* transgenic zebrafish larvae exhibit green fluorescence in neutrophils, which allows a real-time observation of the distribution of neutrophils in zebrafish [21,22]. The number of neutrophils infiltrating near lateral line neuromasts provides a measurement of the degree of inflammation produced. Lipopolysaccharide and copper sulfate (CuSO_4_) are potent inducers of inflammation in zebrafish. In this study, the in vivo anti-inflammatory activity of PP7 was investigated using LPS-stimulated zebrafish embryos and a CuSO_4_-wounded transgenic *BACmpx::GFP* zebrafish model. Moreover, the present study was also conducted to investigate the in vivo anti-inflammatory effect of PP7 in acute and chronic models of inflammation in mice, namely xylene-induced ear edema and cotton pellet-induced granuloma tests, respectively. We found that PP7 possessed potent anti-inflammatory activities against inflammation in zebrafish and mice.

## 2. Results

### 2.1. Effects of PP7 on LPS-Induced NO and PGE_2_ Production in RAW264.7 Cells

In order to determine the optimal concentrations of PP7 for the current study, we examined the cytotoxicity of PP7 on RAW264.7 cells firstly. Polyphyllin VII did not show cytotoxicity of RAW264.7 cells up to the concentration of 1.50 μM. Therefore, in all subsequent experiments, we used concentrations of PP7 ranging from 0.75 to 1.50 μM. We evaluated the potential anti-inflammatory effects of PP7 on NO and PGE_2_ production in LPS-stimulated RAW264.7 cells. The LPS treatment markedly increased NO and PGE_2_ production. Polyphyllin VII inhibited the levels of NO production in LPS-stimulated cells in a concentration-dependent manner by 24%, 33%, 46%, and 62% at 0.75, 1.00, 1.25, and 1.50 μM, respectively (Figure 1A). Moreover, PP7 significantly suppressed LPS-induced PGE_2_ production in a dose-dependent manner by 10%, 21%, 31%, and 46% at 0.75, 1.00, 1.25, and 1.50 μM, respectively (Figure 1B). Since PP7 at less than 1.50 μM did not show toxicity to RAW264.7 cells, we deemed that the inhibitory effects of PP7 on LPS-induced NO and PGE_2_ were not attributable to its cytotoxic effect.

To determine whether the inhibitory effects of PP7 on NO and PGE_2_ production were related to the modulation of *iNOS* and *COX-2* expression, Western blotting and qPCR were performed to determine the *iNOS* and *COX-2* protein and mRNA levels. The results indicated that the protein and mRNA expressions of *iNOS* and *COX-2* were significantly increased in LPS-treated RAW264.7 cells. However, PP7 significantly decreased the LPS-stimulated *iNOS* and *COX-2* protein and mRNA expressions in a concentration-dependent manner (Figure 1C–E).

### 2.2. Effects of PP7 on LPS-Induced Pro-Inflammatory Cytokines Protein and mRNA Expression in RAW264.7 Cells

The pro-inflammatory cytokines, such as TNF-α, IL-1β, and IL-6 produced from macrophages, are the main components of inflammation. Therefore, we tested the inhibitory effects of PP7 on the protein and mRNA expression of *TNF-α*, *IL-1β*, and *IL-6* in LPS-stimulated RAW264.7 cells by ELISA and qPCR, respectively. As shown in Figure 2A–C, the production of TNF-α, IL-1β, and IL-6 significantly increased in LPS-treated cells and markedly inhibited by PP7 in a concentration-dependent manner. Consistently, all the mRNA levels of *TNF-α*, *IL-1β*, and *IL-6* were increased by LPS treatment and remarkably decreased in a dose-dependent manner by PP7 treatment (Figure 2D–F).

### 2.3. Effects of PP7 on LPS-Induced NF-κB p65 Nuclear Translocation and IκB-α Phosphorylation and Degradation in RAW264.7 Cells

Nuclear factor-κB (NF-κB) is known to function as a transcriptional factor to regulate pro-inflammatory mediators in activated macrophages [23] and plays pivotal roles in inflammatory diseases and cancers [24,25]. Since activation of NF-κB occurs following the phosphorylation and degradation of IκB-α by LPS, and NF-κB p65 is the major subunit of NF-κB, the effect of PP7 on LPS-induced phosphorylation and degradation of IκB-α protein and translocation of NF-κB p65 was examined by Western blotting analysis. As shown in Figure 3A–D, the results showed that the levels of cytosolic NF-κB p65 decreased while nuclear NF-κB p65 increased markedly after treatment with LPS. Pre-treatment with PP7 attenuated the cytosolic decrease and nuclear increase of NF-κB p65 protein levels stimulated with LPS. Moreover, the phosphorylation of IκB-α was markedly upregulated by LPS treatment, which was accompanied by a reduction of total amount of IκB-α in macrophages, suggesting a degradation of IκB-α after phosphorylation through ubiquitin proteasome system [26]. However, the phosphorylation of IκB-α was significantly inhibited and its degradation was partially reversed by the treatment of PP7 (Figure 3E). These results indicated that IκB-α degradation and translocation of NF-κB p65 were inhibited by PP7 in the stimulated macrophage cells.

### 2.4. Effects of PP7 on LPS-Induced MAPKs Activation and MMP-9 Expression

Mitogen activated protein kinases (MAPKs) play critical roles in the control of cellular responses to cytokines and stresses and are known to be important for the activation of NF-κB [10,27]. As shown in Figure 4, following LPS treatment, the phosphorylation levels of MAPKs were significantly increased, while PP7 suppressed the LPS-induced activation of JNK, ERK, and p38 MAPK in a concentration-dependent manner. These results suggested that inhibition of MAPKs may contribute, at least partially, to the inhibitory effect of PP7 on LPS-stimulated NF-κB activation in RAW264.7 cells.

Lipopolysaccharide-stimulated MMP-9 expression has been implicated in several inflammatory diseases [28]. In various kinds of cells, different stimuli induce MMP-9 expression through the activation of NF-κB and MAPK signaling pathways [11,29]. The effect of PP7 on MMP-9 activation in the presence of LPS was assessed by Western blotting. Our results showed that the incubation with PP7 in the presence of LPS inhibited MMP-9 protein expression in RAW264.7 cells in a dose-dependent manner (Figure 4). These results suggested that PP7 inhibited the expressions of MMP-9 through downregulating NF-κB and MAPK signaling pathways.

### 2.5. Effect of PP7 on Xylene-Induced Ear Edema in Mice

Because PP7 exerted anti-inflammatory activities in macrophages, studies were extended to determine whether PP7 affected inflammation in animal models. Xylene-induced mouse ear edema reflects the edematization during the early stages of acute inflammation [30]. As shown in Table 1, after xylene application, the weight of the mouse ear increased. However, intraperitoneal injection of PP7 (0.5, 1, and 2 mg/kg) significantly inhibited ear edema formation in a concentration-dependent manner and the inhibition rates were 10.6%, 31.6%, and 39.7%, respectively, as compared to the vehicle control. The positive control group, dexamethasone (DEX, 5 mg/kg) significantly inhibited the xylene-induced ear edema by 57.6% when compared to vehicle control. These results demonstrated that PP7 could significantly inhibit the ear edema induced by xylene in mice.

### 2.6. Effect of PP7 on Cotton Pellet-Induced Granuloma in Mice

The cotton pellet granuloma model has been widely employed to assess chronic inflammation [31]. As shown in Table 2, PP7 (0.5, 1, and 2 mg/kg) as well as dexamethasone (DEX) (5 mg/kg) significantly inhibited granuloma formation surrounding the pellets compared with vehicle control group. The inhibitory values for 0.5, 1, and 2 mg/kg of PP7 were 32.61%, 51.78%, and 55.11%, respectively. The inhibitory value for the group treated with DEX (5 mg/kg) was 57.59%. These results indicated that PP7 could remarkably inhibit chronic inflammation.

### 2.7. Anti-Inflammatory and Protective Effects of PP7 in LPS-Induced Inflammation in Zebrafish Embryo

We further examined the anti-inflammatory activity of PP7 in zebrafish inflammation models. As shown in Figure 5A,B, stimulation of the zebrafish embryo with LPS resulted in an enhancement of NO generation. However, co-treatment of the zebrafish embryos with LPS and PP7 (0.13–0.75 μM) decreased NO generation in a dose-dependent manner. These results demonstrated that PP7 markedly attenuated the increase in NO levels in zebrafish embryos induced by LPS.

Previous studies revealed that LPS-induced inflammation lead to remarkable damage to zebrafish, including small head, tail bending, swelling of the yolk, and an increase of heart-beating rate [32]. To determine the protective effects of PP7 on the toxicity of LPS, we examined the heart beating rate and yolk sac edema size of zebrafish embryos. After the treatment of LPS, the heart beat rate of zebrafish embryo was increased. However, PP7 effectively reduced the heart beat rate to normal level (Figure 5C). In addition, LPS treatment significantly increased the yolk sac edema of the embryos, while PP7 dose-dependently reduced the size of yolk sac edema (Figure 5D). These results indicated that PP7 possesses a profound protective effect against LPS-induced toxicity in zebrafish embryos.

### 2.8. PP7 Inhibits CuSO_4_-Induced Inflammation in Zebrafish

Due to the availability of transgenic lines expressing fluorescent proteins in neutrophils and the optical clarity of zebrafish larvae, zebrafish offer the unique advantage of monitoring an acute inflammatory response in vivo. Copper sulfate (CuSO_4_) is a chemical inducer of inflammation. Temporary treatment of zebrafish larvae with CuSO_4_ selectively triggers cell death of hair cells of the lateral line system and causes rapid neutrophil recruitment to injured neuromasts, leading to a robust acute inflammatory response [33]. In this study, we employed the transgenic *BACmpx::GFP* zebrafish model to evaluate the effect of PP7 on CuSO_4_-induced inflammatory response. As shown in Figure 6, in control fish, neutrophils are localized to the caudal hematopoietic tissue in the ventral trunk and tail. In contrast, in fish treated with CuSO_4_, neutrophils migrate to the horizontal midline and form clusters near lateral line neuromasts. However, pretreatment with indicated concentrations of PP7 inhibited neutrophil recruitment and damage of lateral line neuromasts. These results further confirmed that PP7 is a potent anti-inflammation agent.

## 3. Discussion

Polyphyllin VII (PP7), an active pennogenyl saponin derived from *Rhizoma Paridis*, has been found to exert potent anticancer effects on a wide spectrum of cancer cell lines via suppression of cell proliferation, cell cycle arrest, and induction of apoptosis and autophage [14,15,34,35,36]. In this study, we firstly demonstrated that PP7 exhibits anti-inflammatory activities in vitro and in vivo via inhibiting the production of pro-inflammatory mediators (e.g., NO and PGE_2_) and cytokines (e.g., TNF-α, IL-1β, and IL-6) through downregulating NF-κB and MAPK pathways.

Inflammation is essential for physiological responses to tissue injury and infection. Macrophages actively participate in inflammatory responses by releasing pro-inflammatory mediators and cytokines such as NO, PGE_2_, TNF-α, IL-1β, and IL-6 [37]. Lipopolysaccharide can directly activate macrophages to stimulate the production of these inflammatory molecules [2,3,4]. Nitric oxide is a free radical that plays a pivotal role in many body functions and plays various pro-inflammatory effects on many cell types [5]. However, high levels of NO generated by iNOS in macrophages, in particular, lead to inflammation, cytotoxicity, and autoimmune disorders [38]. Prostaglandin E2 is an inflammatory mediator produced at inflammatory sites by COX-2 and contributes to the development of many chronic inflammatory diseases [6]. Tumor necrosis factor-α, IL-1β, and IL-6 are major pro-inflammatory cytokines in various immune cells and have various pro-inflammatory effects in chronic inflammatory diseases [7]. Tumor necrosis factor-α is a potent activator of macrophages and can upregulate other pro-inflammatory cytokines and endothelial adhesion molecules, which result in the recruitment of neutrophils to the site of inflammation [39]. Interleukin -1β is primarily released by LPS-activated macrophages and plays an important role in the pathophysiology of rheumatoid arthritis [40]. Interleukin -6 is regarded as an endogenous mediator of LPS-induced fever and plays a crucial role in the immune response [41]. Accumulating evidence has shown that overproduction of these inflammatory mediators and cytokines is closely related to inflammatory diseases. Therefore, suppression of these inflammatory molecules’ production provides a promising strategy for the development of anti-inflammatory agents. In the present study, our data showed that PP7 significantly inhibited NO and PGE_2_ production (Figure 1A,B) as well as the protein and mRNA expressions of *iNOS* and *COX-2* in LPS-stimulated RAW264.7 macrophages (Figure 1C–E). Meanwhile, PP7 markedly suppressed the protein and mRNA expression levels of pro-inflammatory cytokine TNF-α, IL-1β, and IL-6 in RAW264.7 macrophages (Figure 2). These results demonstrated that the inhibitory activity of PP7 against NO and PGE_2_ production induced by LPS is attributable to the suppression of *iNOS* and *COX-2* expression. Pro-inflammatory mediators and cytokines have been shown to enhance cancer survival, proliferation, and angiogenesis [42]. We speculate that the anticancer activity of PP7 might be related to the inhibition of inflammatory mediators and cytokines.

Nuclear factor-κB and MAPKs are activated to modulate several pro-inflammatory molecules following LPS stimulation. NF-κB is a transcriptional factor with an important role in immune and inflammatory responses by increasing the expression of pro-inflammatory cytokines and chemokines, such as iNOS, COX-2, TNF-α, IL-1β, and IL-6 [24,25]. In resting cells, NF-κB is present as an inactive form, bound with IκB-α in the cytoplasm. When cells are stimulated by extracellular pro-inflammatory cytokines or LPS, activated IκB kinase complexes phosphorylate IκB-α which lead to its ubiquitination and degradation. NF-κB is released from binding with IκB-α, and then translocates into the nucleus and binds to the promoters of diverse pro-inflammatory genes, ultimately resulting in gene expression [43,44]. In the present study, we found that pre-treatment with PP7 significantly decreased the phosphorylation and degradation of IκB-α and reduced NF-κB p65 level in the nucleus and enhanced NF-κB p65 level in the cytoplasm in LPS-stimulated RAW264.7 cells (Figure 3). These data suggest that the inhibition of inflammatory mediators and cytokines by PP7 might be mediated by blocking NF-κB pathway.

Mitogen activated protein kinases are serine-threonine protein kinases that regulate various cellular activities and are known to play a critical role in the control of inflammatory and immune responses through the upregulation of pro-inflammatory molecules [10,27]. To demonstrate the involvement of MAPKs in the anti-inflammatory effect of PP7, we examined the phosphorylation of MAPKs including ERK, JNK, and p38, the major components of MAPK signaling pathways. Polyphyllin VII significantly inhibited LPS-induced ERK, JNK, and p38 phosphorylation in RAW264.7 cells (Figure 4A–D), suggesting that inhibitory effect on MAPKs might be involved in the suppression of pro-inflammatory cytokines and mediators by PP7. It has been reported that MAPKs play a critical role in modulating NF-κB pathway [45] and the activation of NF-κB by LPS was partially dependent on the phosphorylation of MAPKs [46,47]. Inhibition of MAPK signaling pathway could block the activation of NF-κB pathway [48]. Therefore, we speculate that the inhibition of NF-κB activation by PP7 was attributable, at least partially, to the inhibition of MAPK pathway. MMPs play an important role in the extracellular matrix degradation and remodeling at the sites of inflammation. Among the MMPs, MMP-9 is increased and activated in many kinds of inflammatory and malignant diseases [28]. Matrix metalloproteinase-9 expression can be induced in macrophages by activation of the NF-κB and MAPK signaling pathways [11,29]. In our results, PP7 significantly inhibited LPS-induced expression of MMP-9 in a dose-dependent manner (Figure 4A,E), which may play an important role in the anti-inflammatory activity of PP7.

Since we found PP7 exhibited potent anti-inflammatory activity in LPS-stimulated RAW264.7 cells, we next examined its activity in in vivo models of inflammation. Xylene-induced ear edema is an acute inflammation mice model with severe vasodilation and edematous changes of skin, which involve inflammatory mediators such as bradykinin, histamine, prostaglandins, and serotonin [30]. Suppression of this inflammatory response is likely an indication of antiphlogistic effect [49]. In the present study, PP7 dose-dependently suppressed the formation of xylene-induced ear edema in mice (Table 1), suggesting that PP7 could potently inhibited acute inflammation in vivo via impeding the production of pro-inflammatory mediators as shown in in vitro data. The method of cotton pellet granuloma has been widely used to evaluate the transudative, exudative, and proliferative components of chronic inflammation and the dry weight of the implanted cotton pellets closely correlates with the amount of granulomatous tissue formed [31]. In our results, PP7 inhibited cotton pellet-induced granuloma formation in a dose-dependent manner in mice (Table 2), indicating that it also inhibited chronic inflammation. This reflected its potential efficacy of inhibiting the synthesis of collagens, mucopolysaccharides, fibrosis, and eventually granulation tissue formation [50]. However, further investigations are required to elucidate the underlying mechanisms involved.

Recent studies reported that zebrafish have been increasingly used to assess the anti-inflammatory effect of natural bioactive components in vivo [18,19,20]. In the present study, we found that PP7 acted as a strong inhibitor of NO production in LPS-stimulated inflammatory zebrafish model (Figure 5A,B). In addition, PP7 significantly reduced the heart-beating rate and yolk sac edema size induced by LPS exposure in zebrafish embryos (Figure 5C,D), suggesting that PP7 confers protective activity against the toxicity produced by LPS. Inflammation is a reaction of the immune system to tissue damage and infection. One of the hallmarks of the innate immune response is infiltration of neutrophils in the damaged tissue [51]. Copper sulfate induced inflammation in the transgenic *BACmpx::GFP* zebrafish provides an effective model for easy quantitative measurement of neutrophil infiltration in damaged neuromasts [51,52]. In the present study, we found that pre-treatment with PP7 significantly decreased the neutrophil recruitment toward injured neuromasts (Figure 6). These results consolidate the potent anti-inflammatory activity of PP7. We speculate that there might be association or crosstalk between the anti-inflammatory and anti-cancer effects of PP7, in which the details need to be further elucidated.

In summary, we demonstrated that PP7 has a strong anti-inflammatory activity in RAW264.7 cells through inhibition of pro-inflammatory mediators (e.g., NO and PGE_2_) and cytokines (e.g., TNF-α, IL-1β, and IL-6) production and inflammation-associated genes (e.g., *iNOS*, *COX-2*, and *MMP-9*) expression by suppressing the NF-κB and MAPK pathways. In addition, PP7 exhibited strong anti-inflammatory properties against xylene-induced acute inflammation and cotton pellet-induced chronic inflammation in mouse models. Polyphyllin VII also possessed anti-inflammatory and protective effects against the inflammation and toxicity induced by LPS and CuSO_4_ exposure in zebrafish models. Taken together, these results indicate that PP7 exhibits potent anti-inflammatory activities in vitro and in vivo and suggest its potential application in development of novel therapeutic agents for inflammatory diseases.

## 4. Materials and Methods

### 4.1. Chemicals and Reagents

Polyphyllin VII (PP7) was kindly provided by Prof. Zhinan Mei (South-Central University for Nationalities, Wuhan, China). Dulbecco’s modified Eagle’s medium (DMEM), phosphate buffered saline (PBS), penicillin-streptomycin (PS) were obtained from Gibco (Carlsbad, CA, USA). Fetal bovine serum (FBS) was purchased from Invitrogen (Carlsbad, CA, USA). Thiazolyl blue tetrazolium bromide (MTT), dimethyl sulfoxide (DMSO), LPS (*Escherichia coli* strain 055:B5), and tricaine (ethyl 3-aminobenzoate methanesulfonate salt) were obtained from Sigma–Aldrich Co (St. Louis, MO, USA). Primary antibodies against iNOS, COX-2, phosphor-IκB-α, IκB-α, NF-κB p65, phosphor-ERK, ERK, phosphor-JNK, JNK, phosphor-p38, p38, MMP-9, Lamin B, and glyceraldehyde-3-phosphat-e dehydrogenase (GAPDH), and secondary antibodies were purchased from Proteintech (Chicago, IL, USA) or Cell Signaling Technology (Danvers, MA, USA). Nitric oxide assay kit and 3-amino,4-aminomethyl-2′,7′-difluorescein, diacetate (DAF-FM DA) were obtained from Beyotime Institute of Biotechnology (Nanjing, Jiangsu, China). The PGE_2_ enzyme-linked immunosorbent assay (ELISA) kit was purchased from Nanjing Jiancheng Bioengineering Institute (Nanjing, Jiangsu, China). The TNF-α, IL-1β, and IL-6 ELISA kits were obtained from Biolegend (San Diego, CA, USA). Xylene was purchased from Beijing Chemical Works (Beijing, China). Dexamethasone (DEX) was obtained from Henan Runhong Pharmaceutical Co., Ltd (Xinzheng, Henan, China). Ultrapure RNA extraction kit, SuperRT cDNA First Strand Synthesis kit, UltraSYBR Mixture (with ROX Reference Dye) were purchased from CWBIO Co., Ltd (Beijing, China). Copper sulfate was obtained from Merck (Darmstadt, Hessen, Germany).

### 4.2. Cell Culture and Drug Treatments

A murine macrophage cell line RAW264.7 was purchased from the Cell Bank of Shanghai Institute of Biochemistry and Cell Biology, Chinese Academy of Sciences (Shanghai, China). Cells were maintained in DMEM supplemented with 10% (*v*/*v*) heat-inactivated FBS and 1% (*v*/*v*) P/S, in a humidified atmosphere of 5% CO_2_ at 37 °C. For the following in vitro experiments, PP7 powder was dissolved in DMSO to make stock solutions and then freshly diluted in the basal medium. Moreover, the final concentration of DMSO in RAW264.7 cells treated with different concentrations of PP7 was less than 0.1%.

### 4.3. Cell Viability Assay

Cell viability was measured using the thiazolyl blue tetrazolium bromide (MTT) method [53]. Briefly, RAW264.7 cells (1 × 10^4^ cells/well) were treated with increasing concentrations of PP7 in the presence of 1 µg/mL LPS for 24 h in 96-well plates. Then, the treated cells were washed twice with PBS and incubated in 0.50 mg/mL MTT solution for 4 h at 37 °C. The medium was replaced with DMSO to dissolve the formazan crystals. The absorbance at 570 nm was read by using a microplate reader (BioTek, Winooski, VT, USA), and the cell viability of the control group was set at 100%.

### 4.4. Determination of NO, PGE_2_, TNF-α, IL-1β, and IL-6 Production

RAW264.7 cells (1 × 10^4^ cells/well) were plated in 96-well plates for 24 h and then incubated with PP7 in the absence or presence of 1 µg/mL LPS for 24 h. The culture media were collected for the NO, PGE_2_, TNF-α, IL-1β, and IL-6 assays.

The level of nitrite, the stable end product of NO, in the cultured media was measured as described previously [54]. In brief, the cell culture medium was mixed with an equal volume of Griess reagent. Absorbance was measured at 540 nm after incubation at room temperature for 10 min. PGE_2_ secreted into the medium was measured using a PGE_2_ ELISA assay kit. The concentrations of TNF-α, IL-1β, and IL-6 in the cell culture media were determined by ELISA kits according to the manufacturers’ instructions.

### 4.5. Preparation of Cytosolic and Nuclear Extracts for NF-κB Detection

For isolation of cytoplasmic fractions, RAW264.7 cells were washed twice with ice-cold PBS and resuspended in cytoplasmic extraction reagent (Boster Biological Technology, Wuhan, Hubei, China) on ice for 20 min. Then the supernatants were collected as the cytoplasmic fractions after centrifugation at 14,000× *g* for 10 min at 4 °C. To prepare nuclear fractions, the pellets were resuspended in nuclear extraction reagent (Boster Biological Technology) on ice for 20 min, vortex-mixed and centrifuged at 14,000× *g* for 10 min at 4 °C, the supernatants were collected as the nuclear fractions.

### 4.6. Quantitative Reverse Transcription Polymerase Chain Reaction (qPCR) Analysis

The total RNA obtained from the RAW264.7 cells was isolated using an ultrapure RNA extraction kit. Total RNA (1 µg) was reverse-transcribed using a Super RT cDNA First Strand Synthesis kit. qPCR was performed using the UltraSYBR Mixture and the primers were used as follows: *iNOS*: 5′-ACATCGACCCGTCCACAGTAT-3′ (sense) and 5′-CAGAGGGGTAGGCTTGTCTC-3′ (antisense); *COX-2*: 5′-CTGGTGCCTGGTCTGATGATGTATG-3′ (sense) and 5′-TCTCCTATGAGTATGAGTCTGCTGGTT-3′ (antisense); *TNF-α*: 5′-CACCACCATCAAGGACTCAA-3′ (sense) and 5′-AGGCAACCTGACCACTCTCC-3′ (antisense); *IL-1β*: 5′-CTTTGAAGTTGACGGACCC-3′ (sense) and 5′-TGAGTGATACTGCCTGCCTG-3′ (antisense); *IL-6*: 5′-GTTCTCTGGGAAATCGTGGA-3′ (sense) and 5′-GGAAATTGGGGTAGGAAGGA-3′ (antisense); *GAPDH*: 5′-ACCCAGAAGACTGTGGATGG-3′ (sense) and 5′-CACATTGGGGGTAGGAACAC-3′ (antisense). The cycling conditions were: hot-start activation, 95 °C for 10 min; denaturation, 40 cycles of 95 °C for 15 s; annealing, 60 °C for 60 s; extension, 72 °C for 30 s. *GAPDH* was used as a reference gene.

### 4.7. Western Blotting

The method of Western blotting was the same as our previous reports [14,15]. Briefly, after required treatments, the total proteins of the cell samples were isolated using radioimmunoprecipitation assay (RIPA) lysis buffer. Equal amounts of total proteins were separated by appropriate SDS-PAGE and transferred to a polyvinylidene fluoride (PVDF) membrane. After blocking with skim milk, the PVDF membrane was incubated with specific primary antibodies followed by incubation with the corresponding secondary antibodies. Protein bands were detected by Bio-Rad ChemiDoc^TM^ (Hercules, CA, USA). GAPDH was used as the internal control.

### 4.8. Animals

Male ICR mice (18–22 g) were purchased from Changchun Yisi Experimental Animal Research Center (Changchun, China). The animals were maintained in a 12 h light/dark cycle at approximately 22 ± 2 °C and 50 ± 10% relative humidity, and provided standard laboratory water and rodent chow ad libitum. All procedures were approved by the Institutional Animal Care and Use Committee of Changchun University of Chinese Medicine (Approval No. CUCM-20170015).

The wild-type AB strain of zebrafish and *BACmpx::GFP* transgenic zebrafish [21] were used in the current study and all animal experiments were conducted following ethical guidelines of Biology Institute of Shandong Academy of Sciences (Approval No. BISD-20180928). Zebrafish maintenance were performed as described previously [55].

### 4.9. Xylene-Induced Ear Edema in Mice

The effect of PP7 on acute topical inflammation was evaluated by the xylene-induced ear edema test as previously described [56]. Male Institute of Cancer Research (ICR) mice were divided into five groups of eight animals each. Mice were injected intraperitoneally with PP7 (0.5, 1, 2 mg/kg), DEX (5 mg/kg) or vehicle (normal saline) daily for seven days before the experiments. One hour after the final injection, each mouse received 10 µL of xylene on the ventral and dorsal sides of the right ear; the left ear was used as control without treatment. One hour after induction of inflammation, the mice were sacrificed by cervical dislocation. Ear biopsies of 7.0 mm in diameter were punched out and weighed. The intensity of ear edema was assessed by the weight difference between the right and left ear biopsies of the same mice.

### 4.10. Cotton Pellet-Induced Granuloma in Mice

The effect of PP7 on chronic inflammation was evaluated using the cotton pellet granuloma test in mice [57]. The mice were divided into five groups, each group consisting of eight animals. Under anesthesia, sterile cotton pellets (5 mg) were subcutaneously implanted in the inguinal region of the mice under sterile conditions. Mice were injected intraperitoneally with PP7 (0.5, 1, 2 mg/kg), DEX (5 mg/kg) or vehicle (normal saline) for seven consecutive days from the day of cotton pellet implantation. One hour after the final administration, the mice were sacrificed and the granuloma tissue was dissected out and dried at 50 °C to constant weight. It was inferred that the increase in dry weight of the pellets was regarded as the measure of granuloma formation.

### 4.11. Measurement of NO Production in Zebrafish

Zebrafish larvae at 8 h post-fertilization (hpf) were transferred to 6-well plates containing 4 mL embryo medium, 15 embryos/well, and then pretreated with or without PP7 for 1 h. To induce inflammation, 5 µg/mL LPS was added to the embryos exposed to PP7 for 16 hpf at 28.5 °C. Then zebrafish embryos were treated with 5 µM DAF-FM DA for 1 h in the dark at 28.5 °C, rinsed, and anesthetized with tricaine, and then observed and photographed using a microscope (Olympus IX53, Tokyo, Japan) with a digital camera (Olympus DP73, Tokyo, Japan). The fluorescence intensity of individual zebrafish larvae was quantified to assess NO production using an ImageJ program.

### 4.12. Neutrophil Migration Assay in Zebrafish

*BACmpx::GFP* transgenic zebrafish larvae at 3 days post-fertilization (dpf) were treated with or without indicated concentrations of PP7 for 1 h prior to 20 μM CuSO_4_ treatment for 1 h and were monitored for neutrophils present at the myoseptum, which provides a measure of the degree of inflammation produced. The extent of infiltration of neutrophils is quantified by counting the number of labeled cells detected in the area of myoseptum. All experiments were carried out with a minimum of 15 larvae per condition.

### 4.13. Statistical Analysis

All the data were given as the means ± standard deviation of three independent experiments. One-way analysis of variance analysis with Tukey’s multiple comparison are used in the GraphPad Prism software (La Jolla, CA, USA). Values of *p* < 0.05 were considered as significant.

## Figures and Tables

**Figure 1 molecules-24-00875-f001:**
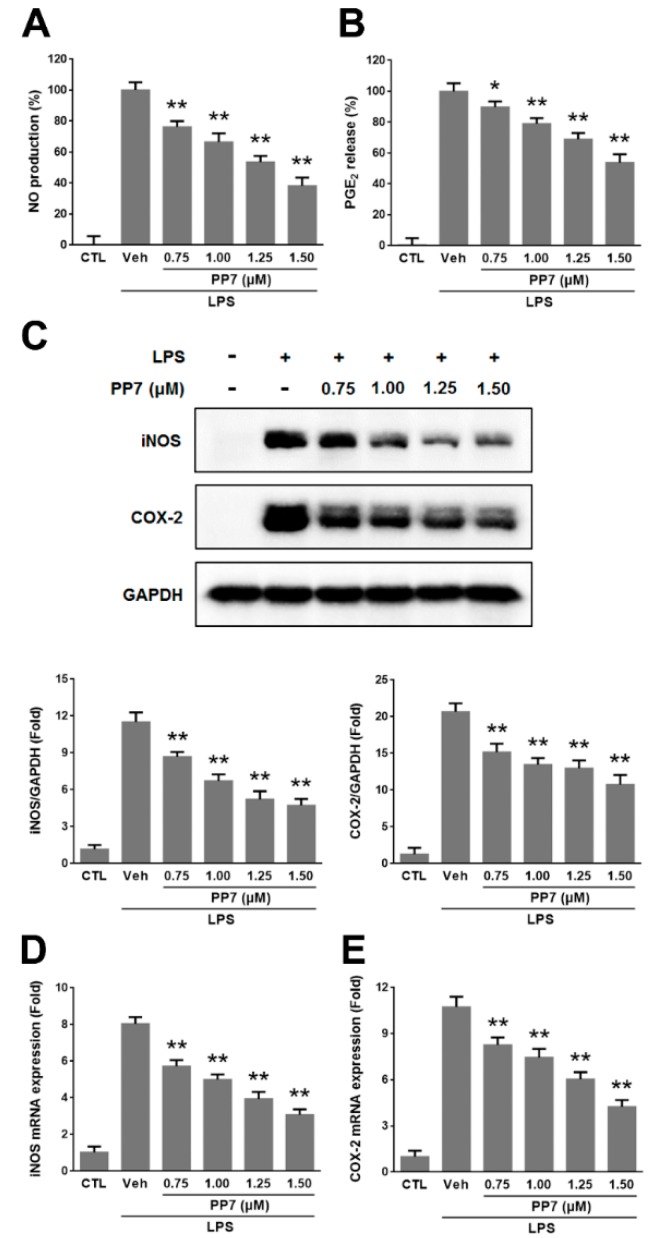
The effects of Polyphyllin VII (PP7) on lipopolysaccharide (LPS)-induced nitric oxid (NO) production (**A**), prostaglandin E2 (PGE_2_) release (**B**), inducible NO synthase (*iNOS*) and cyclooxygenase-2 (*COX-2*) protein and mRNA expression were evaluated in RAW264.7 cells. RAW264.7 cells were pre-treated with different concentrations (0.75, 1.00, 1.25, and 1.50 μM) of PP7 for 1 h and then incubated with or without 1 μg/mL LPS for a further 24 h. NO and PGE_2_ production in the medium was determined by using Griess reagent and enzyme immunoassay, respectively. iNOS and COX-2 protein levels were determined via Western blotting (**C**). The mRNA levels of *iNOS* and *COX-2* were determined by qPCR (**D**,**E**). Values represent the means ± SD of at least three independent experiments. * *p* < 0.05 and ** *p* < 0.01 were compared with LPS-alone group.

**Figure 2 molecules-24-00875-f002:**
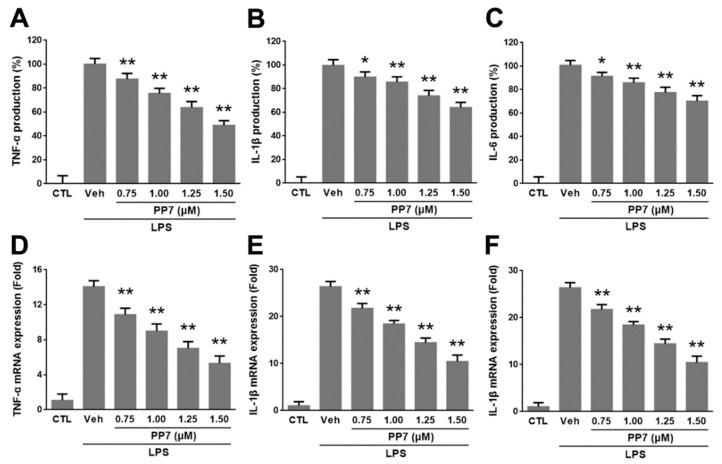
Effect of PP7 on the LPS-induced pro-inflammatory cytokines protein and mRNA expression in RAW264.7 cells. RAW264.7 cells were pretreated with different concentrations (0.75, 1.00, 1.25, and 1.50 μM) of PP7 for 1 h and then incubated with or without 1 μg/mL LPS for a further 24 h. Supernatants were collected, and the tumor necrosis factor (TNF)-α (**A**), interleukin (IL)-1β (**B**), and IL-6 (**C**) production in the supernatants were determined by ELISA. The mRNA levels of *TNF-α* (**D**), *IL-1β* (**E**), and *IL-6* (**F**) were determined by RT-PCR. Values represent the means ± SD of triplicate experiments. * *p* < 0.05 and ** *p* < 0.01 were compared with LPS-alone group.

**Figure 3 molecules-24-00875-f003:**
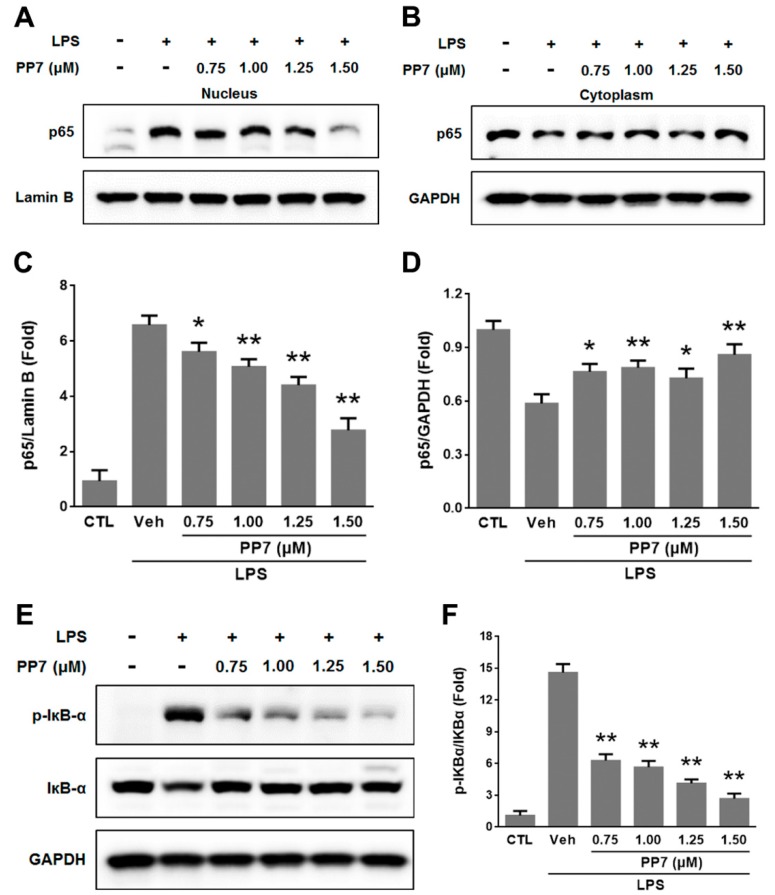
PP7 suppresses LPS-induced NF-κB p65 nuclear translocation and IκB-α degradation and phosphorylation in RAW264.7 cells. Cells were pre-treated with or without different concentrations (0.75, 1.00, 1.25, and 1.50 μM) of PP7 for 1 h, LPS (1 μg/mL) was then added and cells were incubated for 30 min. Western blot analysis for NF-κB p65 in the nuclear (**A**) and cytoplasmic (**B**) extracts of RAW264.7 cells. (**C**,**D**) were densitometric analysis of (**A**,**B**). The levels of p-IκB-α and total IκB-α in the cytoplasm of RAW264.7 cells were determined by Western blotting (**E**,**F**). Values are expressed as means ± SD of triplicate experiments. * *p* < 0.05 and ** *p* < 0.01 were compared with LPS-alone group.

**Figure 4 molecules-24-00875-f004:**
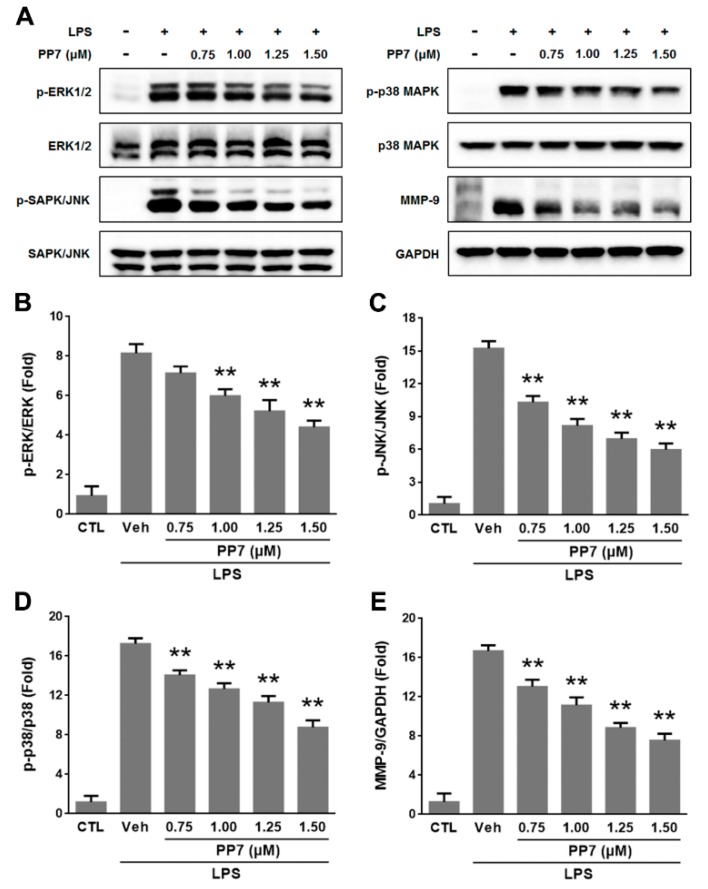
Inhibitory effect of PP7 on the LPS-induced MAPKs and MMP-9 proteins in RAW264.7 cells. Cells were pre-treated for 1 h with or without PP7 at indicated concentrations (0.75, 1.00, 1.25, and 1.50 μM) and then stimulated with LPS (1 μg/mL) for 15 min (**B**–**D**) or 24 h (**E**). (**A**) The levels of p-ERK, ERK, p-JNK, JNK, p-p38, p38, and MMP-9 were determined by Western blotting. (**B**–**E**) were densitometric analysis of (**A**). Values are expressed as means ± SD of triplicate experiments. ** *p* < 0.01 were compared with LPS-alone group.

**Figure 5 molecules-24-00875-f005:**
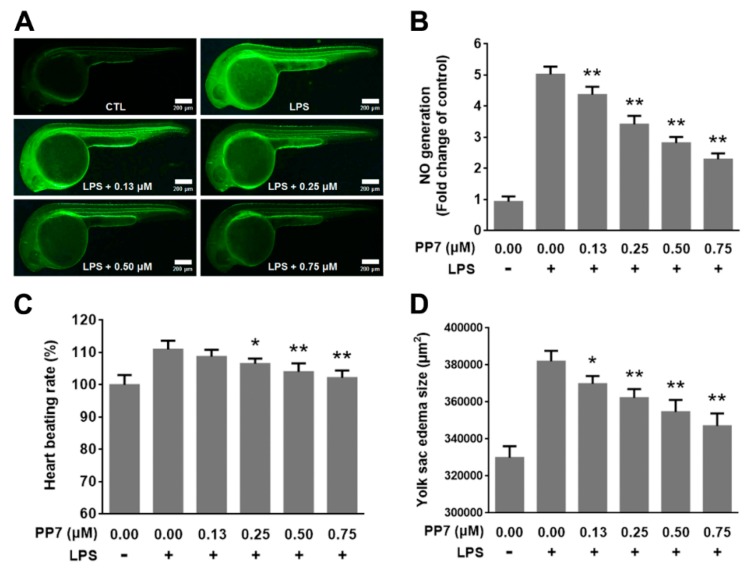
Anti-inflammatory and protective effects of PP7 in LPS-induced inflammation in zebrafish embryo. Zebrafish embryos at 8 h post-fertilization (hpf) were pretreated with PP7 and exposed to LPS. (**A**) The NO generation in zebrafish larvae was measured by image analysis and fluorescence microscope after staining with 4-amino-5-methylamino-2’,7’-difluorofluorescein diacetate (DAF-FM-DA). Scale bars represent 200 μm. (**B**) Quantitative analysis of fluorescence intensity of (**A**) using an image J program. Effects of PP7 on heart beat rate (**C**) and yolk sac edema size (**D**) of LPS treated zebrafish embryos. Data are represented as means ± SD from three independent experiments. *n* = 10/group, * *p* < 0.05, ** *p* < 0.01 as compared with LPS-treated alone.

**Figure 6 molecules-24-00875-f006:**
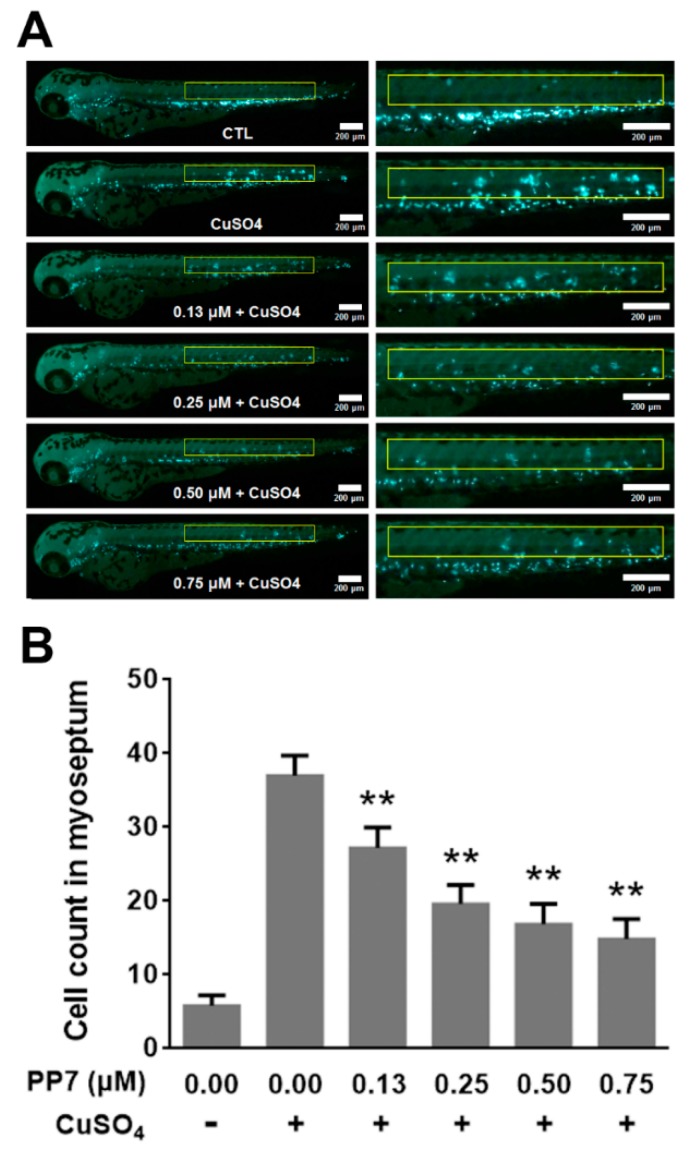
PP7 affects CuSO_4_ induced inflammation in zebrafish. (**A**) *BACmpx::GFP* transgenic zebrafish larvae at 3 days post-fertilization (dpf) were treated with indicated concentrations of PP7 for 1 h prior to CuSO_4_ treatment (20 μM CuSO_4_ for 1 h) and were monitored for neutrophils present at the myoseptum. Scale bars represent 200 μm. (**B**) Quantification of numbers of neutrophils recruited to the lateral line after CuSO_4_ treatment. Data are represented as means ± SD from three independent experiments. *n* = 10/group, ** *p* < 0.01 as compared with CuSO_4_-treated alone.

**Table 1 molecules-24-00875-t001:** Effect of PP7 on xylene-induced ear edema in mice.

Group	Dose (mg/kg)	Edema Degree (mg)	Inhibition Rate (%)
Control	-	6.26 ± 0.59	-
DEX	5	2.65 ± 0.86 **	57.6
PP7	0.5	5.60 ± 0.25 *	10.6
PP7	1	4.28 ± 0.32 **	31.6
PP7	2	3.78 ± 0.31 **	39.7

Values are means ± SEM of differences in weight between right and left ear of mice (*n* = 6). * *p* < 0.05, ** *p* < 0.01 compared with control. Mice were i.p. injected with PP7, dexamethasone (DEX) or vehicle (Control) daily for seven days before inducing ear edema.

**Table 2 molecules-24-00875-t002:** Effect of PP7 on cotton pellet-induced granuloma in mice.

Group	Dose (mg/kg)	Granuloma Dry Weight (mg)	Inhibition Rate (%)
Control	-	28.69 ± 3.23	-
DEX	5	12.17 ± 2.30 **	57.59
PP7	0.5	19.33 ± 2.80 **	32.61
PP7	1	13.83 ± 2.52 **	51.78
PP7	2	12.88 ± 2.31 **	55.11

Values are means ± SEM (*n* = 6). ** *p* < 0.01 compared with control. Mice were i.p. injected with PP7, DEX or vehicle (Control) for seven consecutive days from the day of cotton pellet implantation.

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
