# Peer review of "In Vitro and In Vivo Anti-Inflammatory Effects of Polyphyllin VII through Downregulating MAPK and NF-κB Pathways"

_molecules, 2019, doi:10.3390/molecules24050875_

Round 1

Reviewer 1 Report

In this study, the anti-inflammatory property of polyphyllin VII was evaluated. The experiment was carefully conducted and the results were clearly presented. I think that this research should be published on molecules with minor revision as below,

Please add a figure which indicates chemical structure of polyphyllin VII

Author Response

The reviewer’s comments and suggestions:

In this study, the anti-inflammatory property of polyphyllin VII was evaluated. The experiment was carefully conducted and the results were clearly presented. I think that this research should be published on molecules with minor revision as below,

Please add a figure which indicates chemical structure of polyphyllin VII.

Response: Thanks very much for your encouraging comment. We highly appreciate. We have added the chemical structure of polyphyllin VII in the graphical abstract of this paper according to your useful advice.

Reviewer 2 Report

The article by Zhang C et al. is well structured, very interesting and well written. They showed new data concerning in vitro and in vivo anti-inflammatory effects of Polyphyllin VII (PP7), a steroidal saponin from Paris polyphylla. Previously it has been shown that the PP7 compound has a strong anti-cancer activity. This study focused on investigating the various anti-inflammatory properties of PP7. These activities and its underlying mechanisms of PP7 were evaluated in LPS-stimulated RAW264.7 cells and in mouse and zebrafish models. They showed that PP7 suppressed the LPS-induced NO and PGE2 production and the protein and mRNA expression of iNOS and COX-2 in RAW264.7 cells. Also, they demonstrated that PP7 inhibited TNF-α, IL-1β, IL-6 and MMP-9 by inhibiting NF-κB and MAPKs pathways.

Minor concerns:

1) Concerning results on the inhibition of MMP-9 protein expression in RAW264.7 cells by the incubation of PP7 in the presence of LPS, it would be interesting to show also results with only PP7 compound (Fig. 4).

2) In figure 4 it is not clear where LPS treatment was performed (1 μg / mL) for 15 min or 24 h in which graph B) or C)?

3) In the materials and methods the authors mention that the method of Western blotting was the same as their previous reports. However, they should say in an essential way the important points of the method and the related controls.

4) It would be interesting to know some effects of PP7 given at the same time as the LPS and also at a later time. Why was a kinetic analysis of the PP7 compound not taken into account?

5) In vivo models in the mouse it is important to mention at what time the PP7 treatment was done in figure lgends.

6) PP7 was kindly provided by Prof. Zhinan Mei from South-Central University for Nationalities, but the authors must say in which city.

Author Response

The reviewer’s comments and suggestions:

The article by Zhang C et al. is well structured, very interesting and well written. They showed new data concerning in vitro and in vivo anti-inflammatory effects of Polyphyllin VII (PP7), a steroidal saponin from Paris polyphylla. Previously it has been shown that the PP7 compound has a strong anti-cancer activity. This study focused on investigating the various anti-inflammatory properties of PP7. These activities and its underlying mechanisms of PP7 were evaluated in LPS-stimulated RAW264.7 cells and in mouse and zebrafish models. They showed that PP7 suppressed the LPS-induced NO and PGE2 production and the protein and mRNA expression of iNOS and COX-2 in RAW264.7 cells. Also, they demonstrated that PP7 inhibited TNF-α, IL-1β, IL-6 and MMP-9 by inhibiting NF-κB and MAPKs pathways.

Minor concerns:

1) Concerning results on the inhibition of MMP-9 protein expression in RAW264.7 cells by the incubation of PP7 in the presence of LPS, it would be interesting to show also results with only PP7 compound (Fig. 4).

Response: Thanks very much for your positive comment and your suggestion. But we are sorry that we did not observe the effect of PP7 alone on MMP-9 expression in quiescent RAW264.7 cells (without LPS treatment). As shown in Fig 4, MMP-9 expression in quiescent RAW264.7 cells is very low. We assume that it is hard to observe an inhibitory effect of PP7 on the trace expression of MMP-9 in quiescent RAW264.7 cells. In addition, this is not a parameter of observation in the current study as we focus on the anti-inflammatory effects of PP7, in which LPS-stimulated RAW264.7 cells, a typical in vitro inflammation model, was used.

2) In figure 4 it is not clear where LPS treatment was performed (1 μg/mL) for 15 min or 24 h in which graph B) or C)?

Response: Thank you for the suggestion.

The manuscript has been revised accordingly.

3) In the materials and methods the authors mention that the method of Western blotting was the same as their previous reports. However, they should say in an essential way the important points of the method and the related controls.

Response: Thank you for the suggestion.

The manuscript has been revised accordingly.

4) It would be interesting to know some effects of PP7 given at the same time as the LPS and also at a later time. Why was a kinetic analysis of the PP7 compound not taken into account?

Response: Thank you for the suggestion. In the current study, we followed the standard protocols as reported in previous prestigious papers to establish in vitro and in vivo inflammation models and observe the anti-inflammatory effects of PP7. To make our data comparable with previously reported data, it is not appropriate to change the experimental protocols. We will take the kinetic analysis of the PP7 compound into account to study the anti-inflammatory effects of PP7 given at the same time as the LPS and also at a later time in our future study.

5) In vivo models in the mouse it is important to mention at what time the PP7 treatment was done in figure legends.

Response: Thank you for the comment and suggestion.

The manuscript has been revised accordingly.

6) PP7 was kindly provided by Prof. Zhinan Mei from South-Central University for Nationalities, but the authors must say in which city.

Response: Thank you for the comment and suggestion.

The manuscript has been revised accordingly.